# Consumer Perspectives on Anxiety Management in Australian General Practice

**DOI:** 10.3390/ijerph19095706

**Published:** 2022-05-07

**Authors:** Erin Parker, Michelle Banfield

**Affiliations:** 1Research School of Psychology, Australian National University, Canberra 2601, Australia; 2Centre for Mental Health Research, Australian National University, Canberra 2601, Australia

**Keywords:** anxiety, consumer perspectives, lived experience, general practice, experiences of care, primary care

## Abstract

The aim of the current study was to explore consumer views on the management of anxiety in general practice, which is often the first service from which a consumer seeks professional support. We used a mixed methods survey to explore three broad research questions: (1) what are consumer experiences of anxiety management in general practice, (2) what do consumers prioritise when considering treatment for anxiety and what are their preferences for type of treatment, and (3) how do consumers think care for anxiety could be improved? Consumers reported generally positive views of their GP when seeking help for anxiety, though they had mixed experiences of the approach taken to treatment. Consumers noted that they prioritise effective treatment above other factors and are less concerned with how quickly their treatment works. A preference for psychological intervention or combined treatment with medication was apparent. Consumers noted that key areas for improving care for anxiety were improving access and funding for psychological treatments, increasing community knowledge about anxiety, and reducing stigma.

## 1. Introduction

Anxiety disorders are common in primary care and account for an increasing proportion of the reasons people seek help from a general practitioner (GP) in Australia [1]. To date, much of the research evaluating the management of anxiety and other mental health conditions in primary care has focussed on the description of service data, e.g., [2], the benefits and challenges of providing mental health services in this setting, e.g., [3], and treatment effectiveness according to clinical measures, e.g., [4]. However, the perspectives of people with a lived experience of mental health difficulties (hereafter: *consumers*) are vital in evaluating mental health services. Exploring consumer perspectives is necessary for understanding factors such as barriers in accessing mental health care, priorities for treatment, satisfaction, and areas of unmet need [5,6,7]. Consumers have diverse experiences of care, and understanding their perspectives assists in designing services that more appropriately meet the needs of the people they intend to help [7,8]. In addition to benefits for service evaluation and development, exploring consumer perspectives on mental health care helps identify areas for future research that are most relevant to those consumers, who are the ultimate end-users of health care research [9,10].

A handful of international studies have explored consumer perspectives of primary mental health care, including care for serious mental illness [11,12], experiences of diagnosis for anxiety [13], and expectations for care in anxiety and depression [14]. One recent study explored the quality of care for depression and anxiety in North American integrated primary care settings and found that consumers emphasised the importance of accessibility, good technical care, trusting relationships with providers, and care meeting diverse needs [15]. In Australia, consumer involvement has been a focus of mental health policy since 1992 [16], but research in this area remains sparse and studies that seek to evaluate care from a consumer perspective are few.

This study aimed to explore consumer perspectives on the management of anxiety, specifically in Australian primary care settings. As GPs provide the majority of management for anxiety and are often the first health professional a consumer will see [1,2], we focussed on experiences with a GP specifically. There were three key research questions:What are consumer experiences of anxiety management in general practice?What do consumers prioritise when considering treatment for anxiety and what are their preferences for type of treatment?How do consumers think care for anxiety could be improved?

Participants were also asked about their reasons for help-seeking and any barriers they experienced. We were interested in exploring participants’ first experience of seeking help for anxiety as well as their more recent experiences in the past 12 months.

## 2. Materials and Methods

The ethical aspects of this research were approved by the Australian National University Human Research Ethics Committee (protocol 2019/910). We used a cross-sectional survey to explore consumer experiences and priorities. The survey used a combination of multiple choice, Likert scales, and free response questions and was divided into five broad sections: (1) decision to seek help and expectations, (2) experience and treatment preferences, (3) symptoms and diagnoses, (4) demographics, and (5) overall reflections and suggestions for improvement. The survey was piloted with a small group of people with lived experience from the Consumer and Carer Advisory Group for ACACIA, The Australian Capital Territory (ACT) Consumer and Carer Mental Health Research Unit, who provided feedback on survey content, flow, and length. Adjustments were made to questions following this feedback (i.e., wording and inclusion/exclusion of questions).

### 2.1. Participants and Recruitment

The survey was administered through the Qualtrics online survey platform. Participants were recruited primarily through (1) paid advertisements on the social media platforms Facebook and Instagram, targeted at Australians aged 18 years and older, and (2) consumer peak bodies (Mental Health Australia, National Mental Health Consumer and Carer Forum, Consumer and Community Involvement Program (WA), Flourish (TAS), and ACT Mental Health Consumer Network (ACT)). We also intended to recruit directly from primary health care clinics via flyers placed in waiting rooms. However, due to the COVID-19 pandemic, many clinics had removed reading materials from waiting rooms and, as such, were not accepting advertising material. Large, multi-clinic organisations were contacted via email as an alternative, though only one organisation responded. The survey ran for 12 months (7 July 2020 to 6 July 2021). Multiple rounds of social media advertising were conducted during this period.

Participants were a non-random sample of adult Australians (18 years+) who had sought treatment for anxiety from their GP in the past five years. Those who sought help primarily for posttraumatic stress disorder (PTSD) and obsessive compulsive disorder (OCD) were excluded, as these conditions are no longer categorised as anxiety disorders in current classification systems [17,18]. Participants were assessed as eligible based on their response to screening questions at the beginning of the survey. We intended to recruit a convenience sample of 200 participants in total from all states and territories within Australia to enable detection of small to moderate effects in quantitative analyses and to explore themes among the experiences of a large group of consumers. Participants were not offered incentives for participating in the research. Informed consent was obtained electronically by (a) commencing the survey after reading the participant information sheet and (b) indicating consent for data to be used at the conclusion of the survey by clicking “submit to researcher”. Participants were informed that dropping out prior to submitting their responses would be taken as withdrawal of consent. The survey was administered anonymously, and participants were asked not to enter identifying information in free-response questions. Data were inspected for such information during analysis, and any names of people or specific locations (e.g., GP clinics) were removed.

### 2.2. Survey Measures

The full survey is included in Appendix A. First, participants were asked about their decision to seek help for their anxiety symptoms and whether they were looking for particular treatments, as well as about perceived barriers that prevented them from seeking help. Second, participants were asked about their experience with the care they received, including perceptions of their GP and subsequent treatment approach, satisfaction with care, and perceived effectiveness of treatment. These questions were adapted from existing surveys of health care experience such as the CAHPS^®^ Experience of Care and Health Outcomes (ECHO) Survey [19]. Participant experiences with their GP were assessed using two scales, a seven-item questionnaire exploring perceptions of the knowledge, attitudes, and interpersonal approach of the GP (e.g., “my doctor listened carefully to me” and “my doctor seemed to have good knowledge about anxiety”) and a five-item questionnaire about treatment approach (e.g., “my GP gave me as much information as I wanted about how to manage my anxiety”). Each statement was rated on a five-point Likert scale from 1 “strongly disagree” to 5 “strongly agree”. Participants were asked these questions about their first experience of seeking help as well as recent experiences in the past 12 months.

Information about the location of where participants sought help (remoteness and Australian State or Territory) at both time points was also collected. Participants were also asked whether they had a current mental health treatment plan—a care plan developed with a GP that is required to access subsidised treatment with mental health professional. Each question set ended with a free response question “is there anything else you would like to say about this?” to ensure participants were given an opportunity to provide information they felt was important but may not have been captured by earlier questions.

Information was collected about participants’ demographic characteristics (age, gender, and ethnicity) and known clinical characteristics (lifetime mental health diagnoses and current symptoms). The Depression, Anxiety, and Stress Scale Short Form [DASS-21]; [20] was used to measure anxiety symptoms. The DASS-21 is a self-report, non-diagnostic tool that measures the frequency of depression, anxiety, and stress symptoms experienced over the past week. This measure was chosen over other measures of anxiety symptomatology as it is used frequently in Australian primary care as a screening and assessment tool and can provide information about symptoms that commonly co-occur with anxiety. Participants rate statements on a scale from 0 “never” to 3 “almost always”. Scores are summed within the three subscales with possible scores ranging from 0 to 21 for each subscale. Severity labels (normal, mild, moderate, severe, or extremely severe) are used to classify scores and refer to symptom levels relative to the general population rather than severity of disorder. Scores on the anxiety subscale between 7 and 10 indicate moderate anxiety symptoms, scores between 11 and 13 indicate severe anxiety symptoms, and scores 14 or higher are considered extremely severe. Cut-off scores for severity vary across the subscales. For example, “moderate” refers to scores from 10 to 12 for stress and from 6 to 7 for depression.

The DASS-21 has demonstrated excellent internal consistency with Cronbach’s alphas of 0.94 for the depression subscale, 0.87 for anxiety, and 0.91 for stress [21]. The scales are moderately correlated with one another, consistent with the comorbidity seen in the syndromes they measure (depression-anxiety = 0.42; anxiety-stress = 0.46; depression-stress = 0.39; [20]). However, confirmatory factor analyses with clinical and non-clinical populations have shown that the DASS-21 items can be reliably grouped into the three separate scales [20]. The individual DASS scales also show good convergent and discriminant validity with measures of related and unrelated constructs, respectively [21].

### 2.3. Analysis

Quantitative analyses were conducted using JASP, a free user interface for R available from https://jasp-stats.org/ (accessed on 15 July 2021; [22]). Questions with multiple response selections were divided and coded as 0 (response not selected) or 1 (response selected). We calculated the frequency and proportion of all participants who selected the option as at least one of their responses. The number of responses selected by participants was also calculated for each question.

Participant gender was coded into three groups, male, female, and gender diverse. The overarching category “gender diverse” was used rather than individual categories reported by participants (e.g., non-binary and transgender) to protect confidentiality. Participant ethnicity was coded using the (Australian Standard Classification of Cultural and Ethnic Groups [ASCCEG]; [23]) based on free-response answers from participants. For Likert-scale questions, missing data (n = 3) was imputed to minimise information loss using person-median substitution. Although suboptimal for larger amounts of missing data, this method was considered unlikely to introduce bias in the current study due to the very small number of missing values [24].

In order to explore whether certain variables predicted participant perceptions of their GP, principal components analysis was used as a dimension reduction method for the seven items. A single measure was calculated “perceptions of GP” and scores were compared at first experience and experience in the past 12 months using Wilcoxon signed rank tests. Linear regression was used to explore predictors of participant perceptions of their GP.

Qualitative responses were analysed using content analysis. The analysis used emergent coding, which draws on grounded theory [25] where codes are generated inductively from the data rather than from a pre-existing theory [26]. This process is used to analyse data where research questions are exploratory or broad [26] and was chosen for this study as very little prior research exists regarding consumer views on anxiety care. Participant responses to each question were read for overall understanding by E.P., and open coding was used to generate and assign codes as concepts became apparent. Axial coding was then used to group similar codes into categories. Constant comparative analysis was used throughout the coding process to look at early and later text to ensure consistency in information being recorded by codes and to refine the coding structure. Codes and categories for each question were finalised after no new concepts were identified from the data. To ensure the accuracy of coding and to address potential bias, E.P. discussed the coding structure and key pieces of text with M.B., a lived experience researcher with extensive experience in consumer research and qualitative methods. Following these discussions, refinements were made to the coding structure, including combining codes that reflected similar concepts and separating others that represented distinct concepts.

## 3. Results

A total of 351 people responded to the advertisement and proceeded to the survey on Qualtrics. Of these, 138 completed the survey in full. Participants were advised non-completion of the survey would be interpreted as withdrawal of consent. As such, only complete responses were analysed. A flowchart demonstrating survey response rate can be seen in Figure 1. The median completion time for the survey was 26 min.

### 3.1. Participant Characteristics

The demographic and clinical characteristics of participants are reported in Table 1. Participant ethnicity was classified according to ASCCEG narrow groups (e.g., Eastern European) as many participants did not report their specific cultural and ethnic group (e.g., Hungarian). The majority of participants were female and White, and the median age bracket was 35–44 years. Most commonly reported lifetime diagnoses included generalised anxiety disorder, followed by major depressive disorder. Participants reporting their diagnosis as “other” mostly listed unspecified anxiety or mixed anxiety/depression. Most participants reported having more than one lifetime diagnosis (median = 2). Furthermore, most participants (n = 77, 55.8%) reported they had a mental health treatment plan at the time of completing the survey.

Mean scores on the DASS-21 were moderate to severe for the anxiety subscale (M = 7.4, 95% CI = 6.54–8.24). A quarter of participants’ scores fell in the normal range (n = 37, 26.8%), 13.8% (n = 19) were classified as mild, 9.4% (n = 13) were classified as moderate, 17.4% were classified as severe (n = 24), and approximately a third was classified as extremely severe (n = 45, 32.6%). Mean scores were in the moderate range for the depression (M = 9.0, 95% CI = 7.85–10.06) and stress (M = 10.7, 95% CI = 9.76–11.54) subscales.

### 3.2. Help-Seeking

Frequencies for the reasons that participants sought help for their anxiety are reported in Table 2. Most participants (n = 123, 89.1%) reported that they sought help due to their symptoms becoming too severe to manage. For 89 people, this was the sole reason they sought help, while 26 reported encouragement from others also prompting their help-seeking. Seven participants stated encouragement from others as their sole reason, while a further four stated finding out where to get help as their sole reason. A minority of participants selected combinations of two other reasons (n = 9, 6.5%), and three participants selected more than two reasons for help-seeking.

Half of the participants (n = 69) reported they had first sought help for anxiety prior to 2015. Three quarters of participants reported they experienced at least one barrier to seeking help for their anxiety. Most reported a single barrier (n = 64, 46.4%), 27 (19.6%) reported two barriers, and 13 (9.4%) reported three or more barriers. The most common barrier reported by participants was being afraid to ask for help (Table 2). Among participants who selected “other”, the three most common responses were past negative experience (n = 7, 5.1% of total), shame or stigma (n = 6, 4.3% of total), and a lack of knowledge about anxiety or treatment options (n = 5, 3.6% of total).

Participants were further asked whether they believed that the COVID-19 pandemic had affected their likelihood of seeking help. While almost half of the participants (n = 67, 48.6%) reported that their likelihood of seeking help was unchanged, 59 participants (42.8%) stated the pandemic had made them more likely to seek help. A small number of participants (n = 12, 8.7%) reported decreased likelihood of help-seeking due to the pandemic.

The survey included separate questions about first experience of help-seeking and experiences in the past 12 months. Quantitative and qualitative findings from these two sections are described separately below.

### 3.3. First Experience

Most participants sought help in urban areas (n = 95, 68.8%) in the south-eastern states of Australia (NSW: n = 42, 30.4%; VIC: n = 29, 21.0%; and ACT: n = 23, 16.7%). Twelve participants (8.7%) each first sought help in Queensland and Western Australia, eleven (8.0%) first sought help in South Australia, and eight (5.8%) first sought help in Tasmania. There were no participants with help-seeking experiences in the Northern Territory.

#### 3.3.1. Treatment Preferences—First Experience

Participants were asked whether they had specific treatment preferences at the first appointment with their GP (Table 3). In total, 52 participants (37.7%) reported no preference for treatment at their first appointment (i.e., they had no expectations and/or were looking for general advice), while the majority of participants (n = 86, 62.3%) reported specific treatment preferences. Participants could select more than one response, though most reported a single specific treatment preference (n = 54, 39.1% of total participants). Approximately a fifth of participants reported two preferences (n = 28), and a small number (n = 4, 2.9% of total participants) indicated more than two preferences. Half (n = 69) reported they were seeking psychological treatment via a referral to a psychologist, and approximately a third (n = 43) indicated they were seeking medication. Of the participants seeking medication, most indicated they were also looking for psychological treatment (n = 31, 72.1%). Only 12 participants reported seeking medication alone. By contrast, just over half of the participants (n = 38) seeking referral to a psychologist reported they were looking for this alone.

#### 3.3.2. Treatment Offered—First Experience

Participants were asked which treatments their GP offered at this first appointment. The results are presented in Table 4. Over half of the participants (n = 79, 57.2%) reported at least one of the treatments they were offered was referral to a psychologist. The same number of participants (n = 79, 57.2%) reported being offered medication (short-term medication such as benzodiazepines, long-term medication such as antidepressants or similar, or both). For both treatments (i.e., referral to psychologist or medication), 30 participants (21.7%) reported being offered one but not the other (i.e., medication with no psychologist referral or vice versa). However, most (n = 49, 35.5%) noted being offered both medication and referral to a psychologist. A very small number of participants (n = 3, 2.2%) reported being offered a short-term medication (e.g., benzodiazepines) alone.

Discrepancy scores were calculated for each participant to determine whether there was a difference between preferred treatment and that offered by the GP. For those who had specific preferences (58.7%), most reported they were offered at least one of the treatments they were seeking (n = 62, 44.9% of total participants). Nineteen participants (13.8% of total participants) were not offered any of the treatments they were seeking.

In addition to the type of treatment offered by their GP, participants were asked to rate a series of statements about their GP’s approach to treatment at this first appointment. The results are presented in Figure 2. Similar proportions of participants agreed and disagreed that their doctor gave them information about anxiety (46.7% agreed vs. 41.6% disagreed), gave them treatment options (41.6% agreed vs. 45.3% disagreed), and asked about their preferences (44.5% agreed vs. 46.0% disagreed). When asked to rate whether they received enough information about how to manage anxiety, 38.0% of participants agreed and 52.6% disagreed, with the remainder being neutral. While most participants (54.0%) agreed that they felt able to refuse a specific treatment, more than one-fifth (22.7%) felt they could not refuse. Higher agreement was seen across all items for participants who were offered a treatment consistent with their preferences compared with those who were not (Figure 3).

#### 3.3.3. Perceptions of GP—First Experience

Participants reported generally positive experiences with their GP when they first sought help for anxiety (Figure 4). The highest agreement ratings were for the statement “my doctor showed respect for what I had to say” (70.3% agreed vs. 20.3% disagreed), and the lowest were for “my doctor seemed to have good knowledge about anxiety” (57.2% agreed vs. 23.2% disagreed).

Inspection of the correlation matrix for the seven items regarding perceptions of GP demonstrated correlations of at least 0.65 between all items. The Kaiser–Meyer–Olkin measure of sampling adequacy was 0.88, and Bartlett’s test of sphericity was significant; χ^2^ (21) = 1069.75, *p* < 0.001. Principal component analysis was performed and identified one factor that accounted for 79.6% of the total variance. All items loaded onto the factor at 0.85 or above. A total score for perceptions of GP was therefore calculated (M = 25.96, SD = 8.56) for use in further analyses. Hierarchical linear regression explored the effect of discrepancy in preferred and offered treatment, age, and gender on perceptions of GP. The overall model was not significant when age and gender were included (and neither were significant independently), so they were omitted from the final model. The effect of discrepancy was significant; F(2, 135) = 3.86, *p* = 0.024, R^2^ = 0.054, see Table 5. Perception of GP scores did not vary between participants with no specific treatment preferences and participants who received a treatment consistent with their preferences (t = −0.98, *p* = 0.329). However, treatment being inconsistent with participant preferences was associated with a 6.1 point reduction in ratings of the GP (t = 2.78, *p* = 0.006) compared with preference-consistent treatment. Comparison across individual items (Figure 5) demonstrated particularly low agreement ratings for statements “my doctor spent enough time with me” and “my doctor explained things in a way I could understand”.

#### 3.3.4. Overall Satisfaction and Improvement—First Experience

About two thirds of participants agreed they were satisfied with their experience of seeking help from a GP; 20.3% (n = 28) somewhat agreed, and 38.4% (n = 53) strongly agreed. Just over a quarter of participants reported that they either somewhat disagreed (n = 12, 8.7%) or strongly disagreed (n = 25, 18.1%), and the remainder were neutral. Similarly, 60.6% of participants (n = 83) agreed that their needs were met while 27.7% (n = 38) disagreed. Of the 115 (84.6%) participants who reported receiving at least one of the treatments their GP offered, most somewhat or strongly agreed that it improved their symptoms (n = 77, 67.5%) and quality of life (n = 79, 68.7%).

#### 3.3.5. Qualitative Responses—First Experience

Participants were asked whether they wanted to provide additional information about their first experience seeking help in a free-response question. In total, 64 participants (46.4%) answered this question. Two major themes were identified in the responses: beneficial experiences and adverse experiences.

In total, 25 participants (39.1% of those who provided responses to the open-ended question) mentioned having beneficial experiences with their GP. Many of these participants reported an overall positive experience without detailed information, though ten mentioned their GP being supportive and validating.


*She listened, she took me seriously, she was gentle, and she recommended treatment immediately.*



*I think the best part about seeking help from my GP for the first time was that he listened carefully, was empathetic and validated my experiences. I was so scared before I went in. After telling him about what I was experiencing, I remember him saying “That must be really debilitating for you.” I felt heard and like my problems were real.*


Eight participants also spoke about being satisfied with the approach their GP took to helping them manage their anxiety.


*She didn’t overload me with information that I wasn’t ready for, she just told me the things I needed to know, and what I could handle at that time.*



*It was very positive and her ability to take time to discuss my anxiety with me was really valuable.*


In contrast, 23 participants (35.9% of those who provided responses to the open-ended question) mentioned having adverse experiences when first seeking help from their GP. A major sub-theme among responses was feeling dissatisfied with the treatment or approach the GP took. Ten participants reported this and discussed treatment being inconsistent with their preferences or feeling that they were not given enough information about different treatment options.


*It was a terrible experience and I wish I had had a GP that would have explained my options rather than put me straight on medication.*



*…there was no depth into the symptoms and treatment options. I was given the DASS survey and referred on to a psychologist. It was only when I asked for medication that it was given as a ‘stop gap’. I was given no information on other ways to help with anxiety*


Seven participants also reported that they found their GP dismissive or invalidating.


*She didn’t listen to anything I said. She seemed to be following a script of her own, that was generic and not related to my situation.*



*I was told it was my imagination and I probably just needed a holiday.*



*I was met with complete disregard and my experience belittled. I was told that going outside would be adequate treatment for my crippling fear, which only added to my pain.*


### 3.4. Previous 12 Months

Of the 138 participants, 88 (63.8%) indicated they had seen their GP in the past 12 months for anxiety, not including people who saw a GP for the first time in the past 12 months (n = 23, 16.6%).

#### 3.4.1. Treatment Offered—Previous 12 Months

Almost three quarters of participants (n = 65, 73.9%) who had seen their GP in the past 12 months reported at least one of the treatments they were offered was a referral to a psychologist (see Table 6). In total, 61 participants (69.3%) were offered medication (short term, long term, or both). Most who were offered either medication or psychologist referral were offered both (n = 47, 53.4%), while 18 participants (20.5%) were offered referral to a psychologist with no medication, and 14 (15.9%) reported the opposite. Again, a small number of participants (n = 2, 2.3%) reported being offered a short-term medication alone.

Participants appeared to rate the treatment approach of their GP more highly for experiences in the past 12 months (Figure 6). Agreement ratings were over 50% for most statements. The highest agreement ratings were for feeling able to refuse a specific type of treatment (77.2% of participants agreed). The lowest was for being given information about anxiety (47.7% agreed), though most participants agreed they were given enough information about managing their anxiety.

#### 3.4.2. Perceptions of GP—Previous 12 Months

Participants again indicated positive perceptions of their GP in the past 12 months, with at least 70% of participants somewhat or strongly agreeing with all statements about their experience (Figure 7).

The results of the principal components analysis found the same single factor model for perceptions of GP at 12 months, which accounted for 81.2% of the variance. Composite scores were calculated, and the mean overall score for perceptions of GP was 28.47 (SD = 7.66). The results of the hierarchical linear regression found that the selected participant characteristics (age and gender) and location of help-seeking (urban vs. rural/remote) did not predict perceptions of GP at 12 months; F(9, 77) = 0.65, *p* = 0.755. Due to skewed data, a Wilcoxon signed-rank test was used to compare perceptions of GP at the two time points. The results found significantly higher ratings for perceptions of GP in the past 12 months (M = 28.47) than at first experience (M = 25.96); Z = 2.35, *p* = 0.015, with a moderate effect size of r = 0.35.

#### 3.4.3. Overall Satisfaction and Improvement—Previous 12 Months

Most participants (n = 64, 72.7%) either somewhat or strongly agreed they were satisfied with the experience of seeking help from their GP in the past 12 months, and 73.9% (n = 65) agreed their needs had been met. Of the participants who received at least one of the treatments their GP offered during the past 12 months, most somewhat or strongly agreed it improved their symptoms (n = 60, 73.2%) and quality of life (n = 61, 74.4%).

#### 3.4.4. Qualitative Responses—Previous 12 Months

In total, 50 of the participants who had seen a GP in the past 12 months (56.8%) provided additional information about their experience. As with their first experience, participants’ responses were broadly categorised into beneficial or adverse experiences. Many participants commented they had seen a different GP than at their first experience, which was typically, though not always, related to having a more positive experience. A handful of participants also noted they had first sought help a long time ago and believed that GPs now had improved training and awareness of mental health difficulties.

In total, 18 participants (36.0% of those who provided responses to the open-ended question) provided information about having beneficial experiences with their GP. A major subtheme among these responses (n = 7) was having a caring, supportive, and understanding GP.


*My GP has continued to care for my mental health and anxiety issues, and I feel as though she understands me, and is a partner with me, helping me and guiding me, and willing to listen.*



*My current GP is the perfect example of how a practitioner should treat someone with concerns about anxiety. She listens to me very carefully and is very open and thorough about explaining options.*



*My GP in the last 12 months has always been very caring and has listened well to my concerns about my anxiety. I have no hesitation in approaching him if I needed help/advice.*


A further five participants spoke about their anxiety improving or being resolved.


*I feel so much better and am proud of the progress I have made. I have an appointment every now and then when I want tips/refreshers on managing my anxiety.*


In total, 16 participants (32.0% of those who provided responses to the open-ended question) mentioned adverse experiences with their GP in the past 12 months. This typically related to feeling dismissed by their GP rather than factors related to any treatment offered.


*I feel very rushed and as if my GP just doesn’t have time to see me. She doesn’t take my concerns very seriously anymore…*



*I felt the GPs I consulted were adversely biassed [sic] because of my age, the result was to fail to register the severity of my symptoms.*


### 3.5. Treatment Priorities

Participants were asked to select from a list of factors they thought important when considering treatment for anxiety. Participants could select as many of the options as they wished and were able to include other factors not on the list. Figure 8 reports the percentage of participants who designated the specific factor as important. Almost all participants (n = 127) selected how well the treatment works as important while considering treatment options, while less than half (n = 59) selected how quickly the treatment works. Most participants were concerned with any potential side-effects (n = 91) and factors related to access (e.g., cost: n = 83; ease of access: n = 82). After selecting the important factors, participants were asked to rank their choices from most to least important. The rankings across the five main treatment considerations (i.e., excluding “other”) are presented in Figure 9. Effectiveness of the treatment was the most important factor for most participants (n = 65), followed by cost (n = 32) and potential side effects (n = 25). Only small numbers of participants ranked ease of access (n = 7) or how quickly the treatment works (n = 6) as their top priority.

#### Qualitative Responses—Treatment Priorities

In total, 40 participants (30.0%) provided additional information about their preferences for treatment in a free-response question. Three major themes were identified among the responses: specific treatment preference, problems with treatment, and difficulty accessing treatment.

For example, 17 participants mentioned having a preference for a specific kind of treatment. Among these responses, six discussed a preference for psychological interventions and four expressed they did not want medication without specifying a preference for another kind of intervention.


*These days I prefer psychological treatment above anything else, however I am always open to a medication to help regulate my symptoms, provided the benefits outweigh the side effects.*



*I am not interested in taking medication. I have done so in the past but prefer not to.*


A further four participants discussed a preference for non-clinical or alternative treatments.


*I have had shiatsu massage with mindful meditation as a part of the same treatment. I think there is a wealth of possible treatments that GPs have no idea about.*



*I would really like to get access to ketamine treatment through a psychiatrist as it has been the only effective treatment with no side effects.*


Difficulty accessing treatment was reported by 14 participants, who most commonly spoke about financial cost. For the seven people who mentioned cost as a barrier, this typically related to access to specialist care.


*I would prefer if [p]sychologist visits were better funded by Medicare, both the amount of the rebate and the number of sessions allowed.*



*As a student, cost can be a prohibitive factor for getting help.*


Three people also discussed difficulty accessing treatment due to living in a rural or remote area. Again, this related mainly to specialist mental health care.


*There definitely needs to be better access to mental health services in the country. There are also not enough [p]sychiatrist[s] in the regional areas.*


Nine people spoke about problems with their treatment that were not associated with access issues. Most of these participants discussed concerns about medication side effects and a lack of recognition for this from their treatment providers.


*Very little significance is placed on how the side effects of these medications impact your day to day life. Last time I went on a medication it severely increased my suicidal ideation and reduced impulse control.*


### 3.6. Suggestions for Improvement

At the conclusion of the survey, participants were asked if they had suggestions for improving anxiety care in Australia. In total, 89 participants (64.5%) responded to this question. Four key themes were identified in the responses: better access and funding, improving knowledge and reducing stigma, better training for GPs, and better treatments.

#### 3.6.1. Better Access and Funding

A clear theme in the responses was improving access and funding for mental health services, which was suggested by 34 participants. Typically, this related to access to psychologists although some participants also discussed access to psychiatrists and cheaper medications. Three key sub-themes were identified. First, 15 participants mentioned wanting more affordable options for mental health care generally.


*[D]ecrease costs of treatment—especially psychologists.*



*Make treatment free, I can’t move all the money for treatment around so many times.*



*Cheaper counselling, free medication.*


Furthermore, 13 participants specifically mentioned increasing funding under mental health treatment plans, either through increasing the number of sessions available or increasing the Medicare rebate for services.


*Better Medicare rebates for [p]sychologists, both the amount of the rebate, and the number of sessions allowed.*



*I wish the mental health plan didn’t run out after 10 a year. 10 sessions a year isn’t much when there are 52 weeks of anxiety and depression to get through.*



*Keep the 20 mental health care plan psychology appointments! There have been times in my life I have absolutely needed this and couldnt afford the treatment…*


The third sub-theme was reducing wait-times or increasing numbers of mental health professionals, which was suggested by 11 participants.


*I think more psychologists need to be made available. The wait lists are far too long.*



*Provide affordable support that you do not have to wait months to receive.*


#### 3.6.2. Improving Knowledge and Reducing Stigma

Improving community knowledge about anxiety and reducing stigma was mentioned by 32 participants. The majority of participants (n = 28) discussed increasing general public knowledge about anxiety symptoms, the available treatment options, and how to support those experiencing anxiety.


*More education on recognising the symptoms of anxiety, how common it is, how it can manifest physically. More work to reduce the stigma of anxiety.*



*Teach people how to support someone with anxiety. Education about benzodiazepine use.*



*Greater community awareness, exposure and knowledge about it and its impacts could mean people with anxiety feel less isolated.*


A sub-theme, mentioned by six participants, was improving awareness or reducing stigma specifically in the workplace.


*I think we need to change how mental health is viewed and discussed in the workplace—it is not a personality weakness, it is an illness. Workplaces need to have better processes and attitudes when it comes to managing staff with anxiety or other mental health issues.*



*I think raising community awareness and making workplaces more anxiety friendly will assist in making the path to wellness much more smooth for people living and working with anxiety and other mental health issues in Australia.*


#### 3.6.3. Better Training for GPs

Improving training for GPs was suggested by 15 participants. For many, this related to a need for GPs to have better supportive counselling skills as well as knowledge about anxiety.


*GPs need to have a lot more training in aspects of mental health and ‘listening’ in the doctor-patient relationship.*



*In light of Covid 19 and increased cases of anxiety in the general population, I think it is imperative that GP’s are well-versed in treatment options for anxiety sufferers, how and whom to refer patients onto, and [are] able to [provide] access to concrete information/ie handouts/printouts/phone numbers for patients seeking help for their anxiety.*



*For GPs to be educated more than they currently are about the best first line treatments and how to speak to a patient about their anxiety in a way that is not dismissive.*


#### 3.6.4. Better Treatments

Seven participants also mentioned a need for better treatments for anxiety. Four people discussed wanting better medication options and three discussed wanting alternative treatments to be available.


*More medications which don’t have side effects, or which are anxiety specific.*



*Medicinal cannabis is amazing for anxiety and becoming commonplace in places like the US.*



*Introduce alternative therapies such as kinesiology and aromatherapy.*


## 4. Discussion

This study aimed to explore the experiences and priorities of consumers regarding anxiety care in general practice. Many consumers reported they were initially seeking general advice or information from their GP or had no specific preferences for treatment. However, for consumers who stated an initial treatment preference, it tended to be for referral to a psychologist or combined treatment with medication. Few participants noted preferring medication alone when they first sought help for anxiety. Most participants with specific treatment preferences reported that these were at least partially met.

Overall, participants reported positive perceptions of their GP. Participants indicated they felt listened to and respected, and commented on feeling supported or validated during their interactions with GPs. Qualitative responses tended to emphasise interpersonal aspects of care including among participants who had adverse experiences, noting that this was often due to feeling dismissed or invalidated. This aligns with previous research demonstrating that although consumers want providers with sound clinical knowledge, they value the relational aspects of their mental health care most highly [27,28]. Although perceptions of GPs were positive at first experience, satisfaction with care and the extent to which consumers felt their needs had been met was only moderate. This may be explained by less favourable ratings of the treatment approach taken by the GP at these first experiences, as many people indicated their GP had not asked about their treatment preferences and did not give them enough information about anxiety or treatment options. Qualitative responses echoed this, indicating consumers wanted more in-depth information from their GP to help them understand the different treatment options and make an informed choice.

Participants with unmet treatment preferences had particularly unfavourable perceptions of their GP, and the vast majority indicated their GP had not asked about their treatment preferences. By comparison, almost two thirds of those who received at least one of the treatments they were seeking indicated their GP had asked about treatment preferences. Consumers with unmet treatment preferences also indicated they were generally not given information about different treatment options or enough information about how they could manage their anxiety. Collaborative decision-making is important for consumer experiences of mental health care [29,30], and a lack of ownership over treatment decisions is associated with increased likelihood of disengaging from treatment over time [31]. However, these approaches to care are not yet widely implemented [32], and consumers have reported paternalistic experiences in primary care, where decisions about treatment are made for them rather than with them [11].

Consumers gave more positive ratings of their GP, the treatment approach, and their satisfaction with care in the past 12 months than when thinking about their first experience seeking help for anxiety from a GP. Several participants qualified this by noting they had seen a different GP recently than at their first experience, or perceived care had improved since they first sought help some years ago. Most participants stated their doctor had both asked about treatment preferences and given them enough information about managing anxiety in the past 12 months, compared with less than half of participants who agreed with these statements regarding their first experience. However, it may be the case that participants who had not seen their GP in the past 12 months (n = 50, 36.2%) were more likely to have had negative first experiences and not returned for further care. This may have created a selection bias for people with more positive experiences among those reporting 12 month experiences. Furthermore, half of participants reported that their first experience was more than five years ago, potentially limiting the accuracy of their recollections.

When asked about their priorities for anxiety treatment, consumers reported the most important consideration was effectiveness and were much less concerned with how quickly the treatment works. GPs often report perceptions that consumers expect medications for anxiety and have noted feeling pressure to provide “quick fix” treatments for mental health problems [33,34]. Our results suggest that this may be at odds with the preferences of consumers for anxiety. This may be particularly the case if the trade-off is long-term effectiveness, as in the case of benzodiazepines [35]. The majority of participants also reported potential side-effects as an important consideration in their treatment. This was echoed in qualitative findings, with several participants noting medication side-effects had been an issue with their treatment, and was a factor to consider in improving the care for anxiety. Adverse effects are a key reason consumers cease medication for mental health problems [31,36], and certain side-effects (e.g., sexual dysfunction) and their impact on quality of life are underemphasised in information provided to consumers [37].

Strong themes were identified among responses from consumers about improving care, much of which related to better access and funding for psychological services. This has been noted in previous research on consumer perspectives and is a well-documented issue in the current Australian mental health system [38]. Lack of specialists in regional and remote areas is also a particular concern, which was identified as a thread in participants’ qualitative responses. Many consumers reported barriers to initial help-seeking related to stigma, problems with accessing treatment due to cost, and a lack of knowledge about services. The integration of mental health professionals in primary care is considered imperative in improving mental health care and addressing many of these issues, and trials of such models have been viewed favourably by consumers [15,39]. However, although this is becoming more common in Australia, this is not yet commonplace [3]. Furthermore, while many practices have access to co-located mental health specialists, these are typically privately practicing clinicians working under secondary care referral arrangements [40].

The recently announced permanency of Medicare rebates for telehealth services are important for providing consumers with flexible care and help to address some access and funding issues. However, wait-times for psychologists remain long and the COVID-19 pandemic has resulted in even further demand for services [41]. The finding that the pandemic either did not change or increased the likelihood of seeking help for most participants suggests increased help-seeking among those with existing conditions may account for this increase. Workforce issues are complex to resolve, and rates of anxiety management are increasing in primary care. E-mental health options such as online treatment programs may serve as an appropriate psychological treatment option for many consumers, which circumvents many issues about access and funding [42]. There are many advantages to online interventions, which are available at any time, can be self-paced, and can be used as an adjunct to therapy with a psychologist. Guided versions of these interventions have good evidence and are suitable for GPs to administer in primary care, though purely self-directed programs are also effective [4,43]. However, despite their effectiveness, uptake of these programs and adherence has been relatively low [44]. In line with this, few participants in the study noted being referred to self-help programs by their GP, and online treatments were scarcely mentioned in qualitative responses. Previous research on consumer views has found a preference for face-to-face over e-mental health interventions, though consumers are generally not averse to considering these treatments [45]. There is a perception among the public that e-mental health interventions are less helpful than face-to-face therapy, and professional support has been found to be essential for help-seeking intentions when experiencing psychological distress [46]. Normalising these interventions and emphasising their effectiveness has been found to be important in improving uptake [47], which can be in part facilitated by GPs. Consumers also tend to perceive guided online treatment programs as more acceptable than purely self-guided programs [46], and as such, these programs may be a more appropriate option for treatment at present. That said, GPs also require further education about the effectiveness of online treatment programs and the ways in which they could guide consumers through such programs [48,49].

Finally, participants suggested better education and training for GPs is needed to improve anxiety care more broadly, particularly regarding interpersonal and supportive counselling skills. The evidence for training programs is mixed and tends to focus on improving diagnostic accuracy and clinical treatment practices (e.g., use of medication and referrals) rather than the interpersonal aspects of care. Some research has found that training programs, including brief programs [50], are effective for improving confidence and competence in recognising and managing common mental health conditions [51]. However, other research has found that education programs, on their own, are not sufficient for improving mental health care [52] and are costly to implement on a large scale. Further research is needed to explore the effectiveness of training programs in improving the aspects of care deemed most important to consumers.

### Strengths and Limitations

There has been little prior research exploring consumer views of primary care management of anxiety and almost none in Australia. This study provides important information about consumer experiences and priorities for treatment, which are vital in evaluating and improving anxiety management in Australian primary care. The use of a mixed-methods approach was a strength of the current study, as this provided rich, comprehensive data on the experiences of people with anxiety.

There are also several limitations of this research, primarily regarding the generalisability of the findings due to the use of a non-random sample. Although anxiety is more common in women [53] and women are more likely to seek help [54], women were likely overrepresented in our data. Only a small number of men and an even smaller number of gender diverse people participated, limiting what can be said about their experiences with seeking help. The online nature of our survey and use of social media for advertising means that those with limited access to technology or poorer internet-literacy are unlikely to have participated in the study. People also self-selected into our study after seeing the advertisement, and previous research has demonstrated that people with positive experiences are more likely to respond to surveys about health care satisfaction [55,56]. Furthermore, our survey was cross-sectional and more than half of participants reported their first experience was over five years ago. Due to the retrospective nature of the study, these reflections may be affected by recall bias, and comparisons between consumers first experience and experience in the past 12 months should be interpreted with this in mind.

## 5. Conclusions

This research indicates that consumers perceive interactions with their GP positively when seeking help for anxiety, though they have mixed experiences of the approach taken to treatment. Consumers appear to prioritise effective rather than fast acting treatment and, in many cases, want more information from their GP about how to manage their anxiety. It is important that GPs ask consumers about treatment preferences, as many may come to their first appointment seeking a particular treatment approach and tend to have more negative experiences of care if these expectations are not discussed. Furthermore, it is important to provide information to consumers regarding the different treatment options so they can make informed decisions about their care.

Many consumers appear to prefer psychological interventions and see improving access and funding for these treatments as crucial in improving the care for anxiety in Australia. Raising the profile of e-mental health programs in the community and within primary care may give consumers more options for psychological intervention. Collaboration with consumers to develop information materials for use in primary care may also assist GPs in providing information to consumers about anxiety and the effective treatment options.

## Figures and Tables

**Figure 1 ijerph-19-05706-f001:**
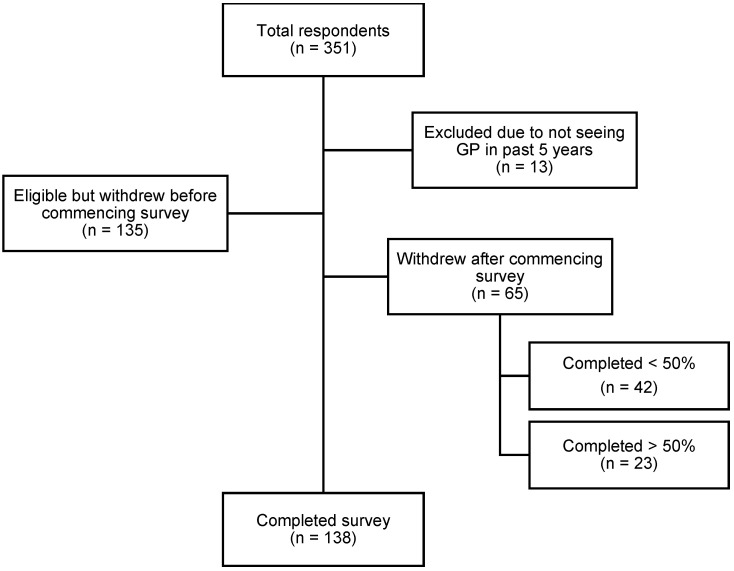
Flowchart demonstrating survey response rate.

**Figure 2 ijerph-19-05706-f002:**
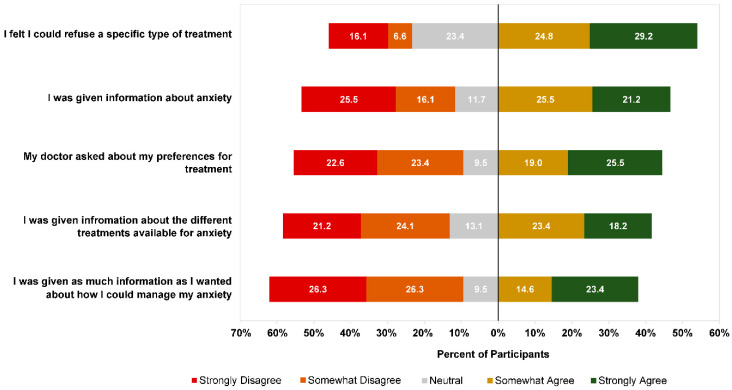
Participant ratings (n = 137) of GP treatment approach at first experience of seeking help for anxiety. Data missing for one participant.

**Figure 3 ijerph-19-05706-f003:**
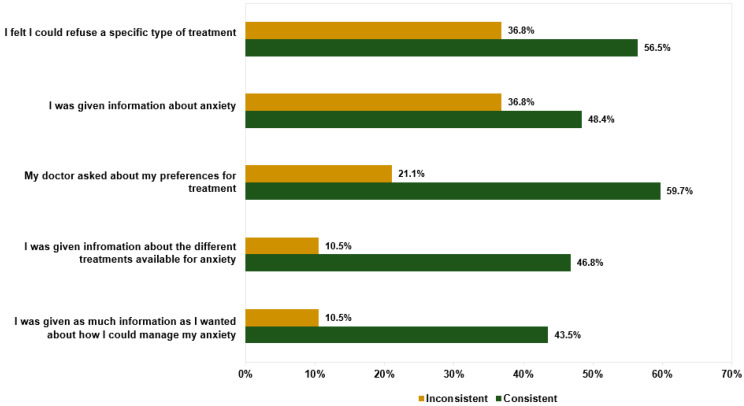
Comparison of agreement ratings with treatment items between participants who received a treatment consistent with their preferences (“consistent”, n = 62) and those who did not (“inconsistent”, n = 19).

**Figure 4 ijerph-19-05706-f004:**
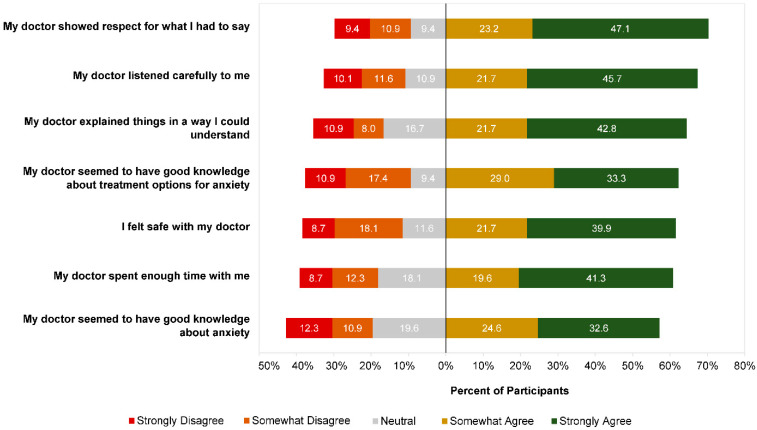
Participant ratings (n = 138) of perceptions of GP at first experience.

**Figure 5 ijerph-19-05706-f005:**
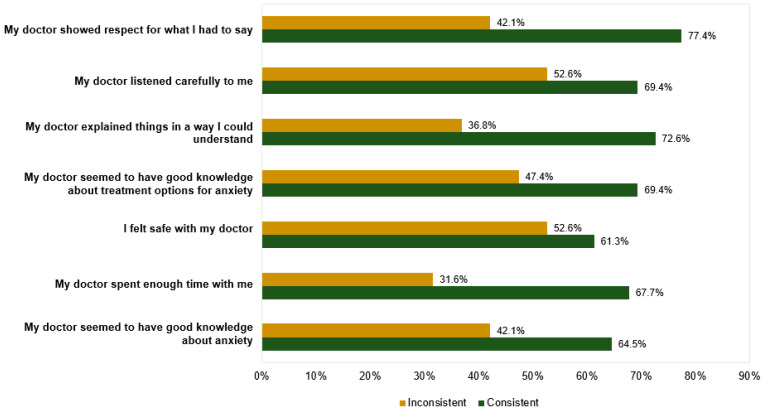
Comparison of agreement regarding perceptions of GP items between participants who received a treatment consistent with their preferences (“consistent”, n = 62) and those who did not (“inconsistent”, n = 19).

**Figure 6 ijerph-19-05706-f006:**
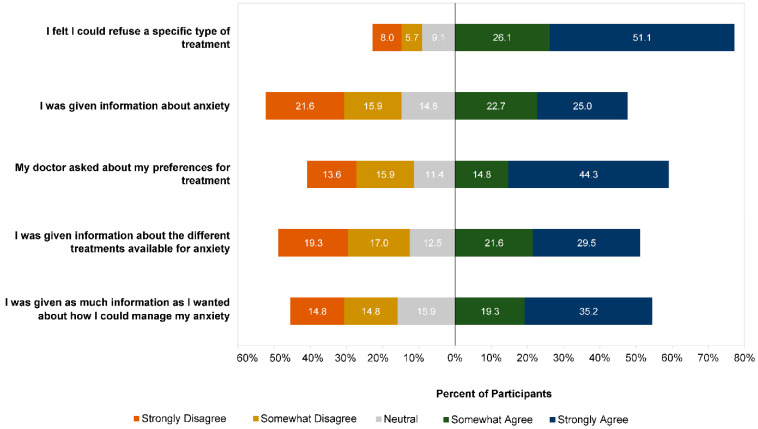
Participant ratings (n = 88) of GP treatment approach in the last 12 months.

**Figure 7 ijerph-19-05706-f007:**
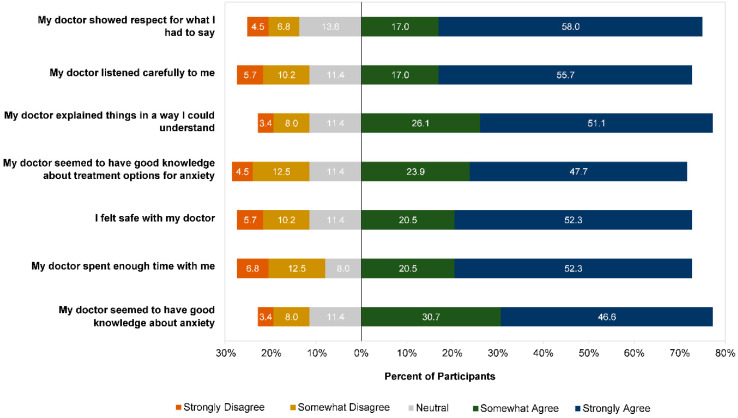
Participant ratings (n = 88) of perceptions of GP in the last 12 months.

**Figure 8 ijerph-19-05706-f008:**
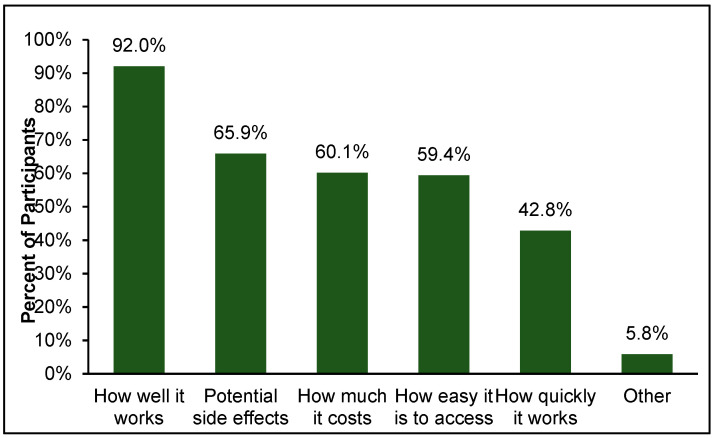
Important factors to participants when considering anxiety treatment.

**Figure 9 ijerph-19-05706-f009:**
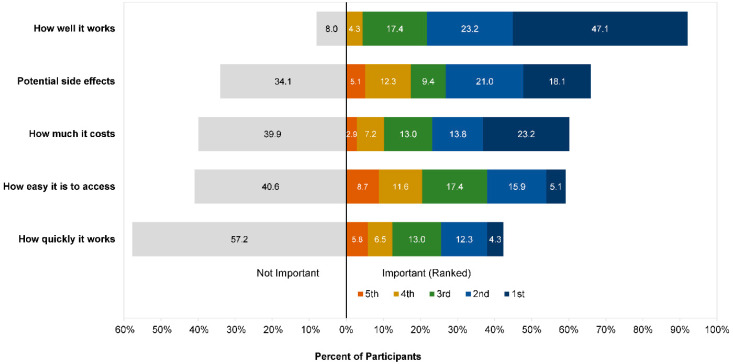
Participant importance rankings for each treatment consideration. “Not important” = percent of participants who did not select the option as important to them, “important” = percent of participants who selected the option as important and the rank they assigned it relative to other considerations (1 = most important to 5 = least important).

**Table 1 ijerph-19-05706-t001:** Characteristics of participants (n = 138).

Characteristic	Variable	Frequency (n)	Proportion of Participants (%)
Age	18–24	15	10.9
	25–34	39	28.3
	35–44	28	20.3
	45–54	22	15.9
	55–64	23	16.7
	65+	11	8.0
Gender	Female	112	81.2
	Male	19	13.8
	Gender diverse	7	5.1
Ethnicity ^a^	Australian	114	83.3
	Aboriginal or Torres Strait Islander	2	1.4
	New Zealander	1	0.7
	British	4	4.3
	Irish	1	0.7
	Western European	1	0.7
	Eastern European	3	2.2
	Chinese Asian	1	0.7
	Southern Asian	3	2.2
	South American	1	0.7
	Multiple	3	2.2
Lifetime diagnosis	Generalised anxiety disorder	78	56.5
	Panic disorder	17	12.3
	Social anxiety disorder	16	11.6
	Specific phobia	9	6.5
	Agoraphobia	6	4.3
	Major depressive disorder	57	41.3
	Other depressive disorder	5	3.6
	Obsessive compulsive disorder	15	10.9
	Posttraumatic stress disorder	35	25.4
	Adjustment disorder	7	5.1
	Bipolar disorder	9	6.5
	Autism spectrum disorder	5	3.6
	Attention deficit hyperactivity disorder	4	2.9
	Schizophrenia spectrum disorder	1	0.7
	Substance use disorder	5	3.6
	Personality disorder	9	6.5
	Eating disorder	13	9.4
	Other	11	8.0
	No diagnosis	20	14.5
	Unsure/prefer not to say	2	1.4
Lifetime diagnoses (n)	0	20	14.5
	1	28	20.3
	2	34	24.6
	3	31	22.5
	4	10	7.2
	5+	13	9.4
First help-seeking (year)	<2015	69	50.0
	2015	9	6.5
	2016	2	1.4
	2017	18	13.0
	2018	11	8.0
	2019	10	7.2
	2020	13	9.4
	Unsure	6	4.3

^a^ Data missing for four participants.

**Table 2 ijerph-19-05706-t002:** Participant reported reasons for and barriers to seeking help for anxiety.

	Frequency (n)	Proportion of Participants (%)
**Reason for help-seeking**		
Symptom severity	123	89.1
Encouragement from others	36	26.1
Found where to go to get help	10	7.2
Other reason	7	5.1
**Barriers ^a^**		
Afraid to ask for help	54	39.1
Financial cost	29	21.0
Unsure where to seek help	24	17.4
Unable to access help	19	13.8
Other	36	26.1

Note. Participants could select more than one response so proportions add to more than 100%. ^a^ Data missing for one participant.

**Table 3 ijerph-19-05706-t003:** Preferences for treatment approach at first appointment with GP.

Treatment Approach	Frequency (n)	Proportion of Participants (%)
Referral for a psychologist	69	50.0
Medication	43	31.2
No specific treatment	34	24.6
General advice	30	21.7
Other	10	7.2

Note. Participants could select more than one response so proportions add to more than 100%.

**Table 4 ijerph-19-05706-t004:** Treatments offered by GP at first appointment.

Treatment Offered	Frequency (n)	Proportion of Participants (%)
Referral—psychologist	79	57.2
Medication—long-term	69	50.0
Lifestyle	60	43.5
Medication—short-term	27	19.6
Counselling by GP	18	13.0
Referral—psychiatrist	16	11.6
Referral—self-help	11	8.0
Other	11	8.0
None	9	6.5

Note. Long-term medication refers to antidepressants or similar, while short-term medication refers to short-acting drugs such as benzodiazepines.

**Table 5 ijerph-19-05706-t005:** Linear regression results for effect of treatment discrepancy on perceptions of GP.

				95% CI		
	Estimate	se	t	LB	UB	*p*
Intercept	27.42	1.07	25.75	25.31	29.52	<0.001
Consistent (reference)	0.00					
Inconsistent	−6.10	2.20	−2.78	−10.45	−1.76	0.006
No Preference	−1.51	1.54	−0.98	−4.55	1.55	0.329

**Table 6 ijerph-19-05706-t006:** Treatments offered by GP in the past 12 months.

Treatment	Frequency (n)	Proportion of Participants (%)
Referral—psychologist	65	73.9
Medication—long-term	56	63.6
Lifestyle	50	56.8
Referral—psychiatrist	24	27.3
Medication—short-term	17	19.3
Counselling by GP	8	9.1
Referral—self-help	6	6.8
Other	3	3.4
None	2	2.3

## Data Availability

All pertinent data reported in published article. Raw data available upon request.

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
