# Peer review of "Consumer Perspectives on Anxiety Management in Australian General Practice"

_ijerph, 2022, doi:10.3390/ijerph19095706_

Round 1

Reviewer 1 Report

I found this an interesting and timely report. It was well written and presented. A similar uk report would provide a similar account of service  users experience of anxiety management a preference for psycho;ogicsl support as well as medication.

Author Response

Thank you for your reviews, we appreciate your time in reading our paper.

Reviewer 2 Report

I enjoyed reviewing this paper. It is very well written and the topic is relevant. It is quite long, but straightforward and comprehensive. The discussion section was relevant. 

Author Response

(The authors gave the same response as above.)

Reviewer 3 Report

Thank you for the opportunity to review this very well written manuscript. It addresses a critical issue and has pragmatic implications for the field.

I am unable to comment on the statistical analyses reported in this paper and do hope that other reviewers with requisite expertise can do so.

Suggest authors consider the statement referring to 'themes' emerged' - in the qualitative inquiry field, recent work (Braun and Clarke) trouble this - they even have buttons with "themes do NOT emerge" written on them - consider using "themes were identified" instead throughout the manuscript

The two themes - positive and negative experiences are put forward - this is quite simplistic - suggest that this is more complex than this basic dichotomisation - were there respondents who indicated both experiences, depending on context? Is there something in vivo that could more powerfully address these areas?

Some subsections have the same heading 'qualitative response' - please address this

Suggest coming back to the issue of GP training in the discussion and draw on the extant literature in this space - why, despite several programs/workshops/initiatives to enhance GP knowledge/awareness/capacity to deal with mental health issues, is there relatively lack of uptake? 
